# Use of a Validated Risk Perception Questionnaire for the Inclusion of People with Hearing Impairments in a Productive Environment

**DOI:** 10.3390/ijerph22060884

**Published:** 2025-05-31

**Authors:** Aline Sias Franchini, Antonio Augusto de Paula Xavier, André Luiz Soares

**Affiliations:** 1Ponta Grossa Campus, Federal Technological University of Paraná, Ponta Grossa 84017-220, PR, Brazil; 2Postgraduate Program in Production Engineering (PPGEP), Federal Technological University of Paraná, Ponta Grossa 84017-220, PR, Brazil; augustox@utfpr.edu.br; 3Câmpus Guarapuava, Federal Technological University of Paraná, Guarapuava 85051-010, PR, Brazil; andresoares@utfpr.edu.br

**Keywords:** well-being, hearing impairment, inclusion, industrial environment, logistics environment, risk perception, ergonomics

## Abstract

The inclusion of people with hearing impairments in logistics environments is a topic of increasing importance, especially when considering the promotion of diversity and accessibility in the workplace. The use of a risk perception questionnaire can represent an effective tool to identify barriers and challenges faced by these workers. The questionnaire covers several aspects, including communication between employees, the adequacy of visual signage, workplace safety and the accessibility of tools and equipment. The responses collected help to understand how people with hearing impairments perceive the risks inherent in their work activities, as well as to identify possible deficiencies in the training and awareness of their coworkers. Analysis of the data collected by the questionnaire allows companies to develop specific strategies and actions to improve inclusion, such as inclusive training, adaptation of the physical environment and implementation of auxiliary technologies. In this way, these actions not only contribute to a safer and more productive work environment, but also value diversity and the unique contribution that each worker can make to the logistics sector.

## 1. Introduction

People with hearing impairments face a number of challenges in the workplace, especially in dynamic sectors such as logistics, where communication is crucial and there is frequent traffic of mobile machines, such as forklifts [1]. The logistics environment is an area that encompasses the management and organization of processes involving the transportation, storage and distribution of products [2].

However, the inclusion of people with hearing impairments in this context is an aspect that deserves special attention, since effective communication is essential for the success of these operations, as is interaction in the constant movement of forklifts in the work environment [3,4].

The safety of logistics workers, especially those with hearing impairments, is an issue of increasing importance in the management of large logistics distribution centers. With the increase in automation and intensive use of machines, it is essential to implement guidelines and policies that ensure the safety and well-being of workers in environments where forklifts are part of the daily routine [3].

In relation to well-being and safety, ergonomics plays an essential role in understanding and improving human interactions with products, equipment, environments and systems [5]. Ergonomics aims to optimize performance, protect health and improve the safety of individuals [5,6].

The main motivation of this article was to contribute to diversity and accessibility in the workplace regarding the inclusion of people with hearing impairments in logistics environments. To this end, the authors developed a risk perception questionnaire that assesses the barriers and challenges faced by employees with hearing impairments in the logistics sector of a company.

The development of a methodology for the inclusion of people with hearing impairments in an environment with machine interactions allows these workers to develop in all work processes [7]. Hearing impaired people should optimally be involved in a more dynamic, safe and stimulating work environment [7,8].

Implementing training programs that address both forklift operation and the specific needs of operators with disabilities, such as sign language training, can significantly reduce these risks [9,10]. Implementing assistive technologies, such as warning devices and visual or audible signals, can improve and strengthen operator safety in environments with moving machines [11]. Inclusive work environments not only improve employee morale but also increase productivity. [12].

## 2. Materials and Methods

### 2.1. Data Collection Procedure

The study was carried out in light of the difficulties encountered by a large company in the automotive sector regarding the inclusion of people with hearing disabilities in industrial logistics processes with safer Human–Machine interactions.

Exploratory research was carried out with the aim of exploring the theme of the study. Descriptive research with a qualitative approach describing the population involved, the place to be studied and the activity to be developed was utilized through bibliographic and systematic literature review with the Methodi Ordinatio [13].

Research planning began with a consultation of the Scopus and Science Direct databases to build the desired portfolio to be studied. The Methodi Ordinatio methodology determines which scientific articles are most relevant within the selected theme [13]. Bibliographic research in the databases was conducted using the keywords that best described the research themes: hearing impairment, inclusion and assistive technologies.

-Filtering procedures:

A search configuration with temporal cuts was utilized, incorporating Boolean operators and searching in the title, abstract and keyword fields. Articles and review articles were selected, and duplicate and off-topic articles were excluded.

At the end of the application of the method, each article received a grade or score, called InOrdinatio, which took into account the following factors: journal impact factor, number of citations of the article and year of publication. The articles with a positive InOrdinatio score, totaling 63 articles, were selected to be read and analyzed [13].

Based on the results of a temporal bibliometry, Figure 1 presents the number of articles published per year that accounted for the study of people with hearing impairments, inclusion and assistive technologies. There has been an increasing trend in the number of publications over the years.

Initially, a questionnaire was applied to the four hearing-impaired workers and four forklift operators who interact with them. The objective of this questionnaire was to collect data to guide the preparation of the risk perception questionnaire and identify the difficulties of this group in their logistics work environment.

The Informed Consent Form (ICF) was read and explained to the participants in oral language and in Sign Language-Libras and/or using a simultaneous sign language translation platform (ICOM), which ensured understanding of the importance of this task. The participants were informed that they could withdraw at any time, that the researcher would clarify any doubts or discomfort generated by the research and that there would be no cost or payment for participation. After acceptance, each participant filled out and signed a form. All procedures were in accordance with ethical principles, and the project was approved by the Ethics Committee, CAAE: 75525323.8.0000.5547.

The hearing-impaired operators that participated in the study had hearing loss that, from a medical point of view, did not require the use of hearing aids. These individuals moved carts, working on the assembly of material kits that went to the production line.

Of the 39 corridors where the kits were assembled, only 7 did not involve human-machine interaction. These are the ones where hearing-impaired people worked, as the safety conditions were better.

Figure 2a shows the location where the study was conducted. The activity of hearing-impaired operators takes place in a corridor where there is no interaction between man and machine, demonstrated in Figure 2b.

The daily activities of hearing-impaired workers were monitored for a period of 2 h twice a week. This monitoring took place in the months of July and August 2024, and the routine activities of hearing-impaired logistics workers and the forklift operators who interact with them were observed.

The objective of this manuscript was to develop an easy-to-understand and easy-to-apply questionnaire on risk perception. To develop the Risk Perception Questionnaire for the hearing-impaired, the most important items regarding the workers’ responses in the data collection questionnaire were considered, such as communication, training and qualifications, machine movement, and safety items, as shown in Figure 3.

### 2.2. Development of Risk Perception Questionnaire

The development of a tool to assess the risk perception of hearing-impaired workers is important, especially when it comes to logistics environments where there is interaction with machines and equipment. The existence of occupational hazards can represent an even greater challenge for these workers, who may not perceive audible signals that alert them to danger, such as alarms and warnings emitted by machines [14,15].

The following steps were employed in the development and validation of this questionnaire:(1)Literature review: Searches were carried out in periodicals, as previously mentioned, and no instrument was found that could provide a basis for this proposed study. There are some scientific studies regarding the development of technologies in order to facilitate the inclusion of hearing-impaired people in their daily lives and not in an industrial environment.(2)Expert Panels: For this stage, a committee was selected that was composed of specialists from different areas of study that were qualified to judge the validity of the instrument. The group included two ergonomists, two occupational health professionals, two occupational safety technicians, two speech therapists and one logistics engineer.

They were invited to participate in the validation process on a voluntary basis and were given the instrument for review, which they could return at their convenience. They were able to provide feedback on the clarity, relevance and appropriateness of the questions. A common method for assessing content validity is peer review, which emphasizes that expert opinion is vital in determining whether questionnaire items are appropriate in terms of clarity, relevance and comprehensiveness [16]. Experts should therefore assess whether each measurement item faithfully reflects the construct it is intended to measure, contributing to the understanding of the phenomenon under study [17].

To validate the instrument, a simple document with an explanatory heading and consisting of a qualitative and quantitative Likert scale (from 0 to 10 points) applied after each of the questions was created. A Likert scale is a rating scale used to measure opinions, attitudes or behaviors [18].

After the Likert scale, there was also a specific field in which the expert could make observations regarding the wording of the questions and suggest changes if he or she noticed any type of comprehension problem. Each evaluator/expert would then rate on the numerical scale according to the validity of each of the questions.

In addition to relevance and clarity, every measure on a proposed instrument must meet two minimum requirements: validity and reliability [19]. Valid measures are those that accurately represent the phenomenon they intend to measure, while reliable measures are those that are consistent across time and space and can be replicated with different possible applicators [19,20].

The results of the relevance test were satisfactory, with low standard deviation and coefficient of variation for all of them. The final average of the instrument in the relevance test was 8.8.

The risk perception tool was proven to be adequate and reliable. The tool can therefore be effective in assessing risk perception and facilitating the inclusion of people with hearing impairments in logistics processes.

(3)Test with a group of hearing-impaired individuals from other sectors: One of the most common methods for assessing the reliability of a research instrument is the test/retest, which involves applying the same questionnaire to the same group of participants at two different times [21]. Five hearing-impaired workers who also work in a production environment were selected to verify the clarity and relevance of each question. The simultaneous sign language translation platform (ICOM) was used to improve the clarity and interpretation of the questions in the questionnaire to be validated.

The Likert Scale was also applied to each worker in this sample. While applying the test in September 2024 and the retest in November 2024, it was observed whether there was variance between the responses made at different times.

The results of the consistency of the responses showed a total variance result for the sets (X) and (S) of 0.8. The test/retest validation obtained a total variance result for sets (X) and (S) of 0.8, indicating a good reliability and consistency of the responses.

(4)Cronbach’s alpha coefficient

Cronbach’s alpha coefficient is a fundamental tool for analyzing the reliability of questionnaires. Interpretation must be conducted in conjunction with other analyses to ensure the robustness and validity of the conclusions obtained from the collected data [22].

Cronbach’s alpha is a statistical measure widely used to assess the internal reliability of scales and questionnaires, especially in research areas such as psychology, social sciences and education [22]. This coefficient allows researchers to check the consistency of participants’ responses across different items in a questionnaire that theoretically measures the same construct or concept.

Cronbach’s alpha ranges from 0 to 1, with higher values indicating greater reliability. Cronbach’s alpha values below 0.6 indicate unacceptable reliability, values between 0.6 and 0.7 are considered acceptable, values between 0.7 and 0.9 are considered good, and values above 0.9 may indicate excessive redundancy among items [22,23].

To calculate Cronbach’s alpha, the questionnaire was separated into two data lists, X and S, and the mean and variance of each data set were calculated. The total variance of sets (X) and (S) was 0.8, a value considered a good result [23]. 

### 2.3. Application of the Risk Perception Questionnaire

After validating the questionnaire, it was applied to 10 people with hearing impairments in the company’s logistics sector, with a larger hiring audience in 2025. The questionnaire consisted of open and closed questions and was divided into items relating to communication, training and qualifications, machine handling, and safety items, subdivided as follows according to the questions:

This questionnaire was applied to hearing-impaired operators who worked in corridors with machine interaction and those who were in the distribution center environment at another address, all of whom currently work in the company’s logistics sector. Care was taken not to disrupt the work routine, and the area leader assisted. The ICOM platform was used to ensure that questions were understood and answered clearly.

## 3. Results

The results are presented in the form of graphs for better understanding. They are divided according to Table 1, shown previously. The analysis of the results is based on Block 3 of the questionnaire. Blocks 1 and 2 are related to general data about the worker.

Block 3—CommunicationBlock 4—TrainingBlock 5—Interaction with machine operatorsBlock 6—Safety items

### 3.1. Block 3—Communication

In Block 3 of the Questionnaire, identified in Figure 4, related to communication, it was identified that 90% of the employees who completed the questionnaire communicate effectively through lip reading and sign language, and 10% communicate in writing. In the company studied, the most common form of communication was through the ICOM platform.

### 3.2. Block 4—Training

As shown in Figure 5, all employees had training in ICOM, which is the platform that was used for all dialogues in the work environment. By using a professional language medium, there is confidence in the information conveyed.

### 3.3. Block 5—Interaction with Machine Operators

Block 5 refers to the perception of risks by people with hearing impairments when interacting with machine operators. Figure 6 addresses the perception of people with hearing impairments when they are close to machine operators in the same location on the same street/corridor where they work.

Insecurity regarding machines in the workplace is a significant concern, especially in sectors such as logistics, where interaction with equipment is frequent. Of the total number of people with hearing impairments who work alongside machine operators, 70% say they feel insecurity/fear, and 30% say they need to be very careful when sharing the same space with the machine operator.

### 3.4. Block 6—Safety Items

At the end of the questionnaire, there was a section for people with hearing impairments to report comments through an open-ended question, as shown in Figure 7. Workers could cite previous experiences as well as difficulties in their work routine.

A total of 40% of employees reported difficulty in seeing the logistics building’s light signals, such as the visual fire warning. A total of 50% of employees reported the need for greater attention from machine operators who circulate near them. Finally, 10% of employees mentioned that the lack of attention was probably due to the fact that some machine operators were not from the same sector as them and did not have knowledge of people with hearing impairments. This is a concern that could be passed on to leaders so that they can check preventive actions.

The data collected in the questionnaire reveal important information about the experience of employees with hearing impairments in relation to safety and the work environment. Future actions can be planned based on these results.

## 4. Discussion

The proposed study presents a relevant contribution to the inclusion of people with hearing impairments in industrial logistics environments, acting as a potential driver for safer and more inclusive practices. Although it was conducted in a large logistics-sector company in the southern region of Brazil, its tools and methodologies can be applied in different locations and sectors, expanding the scope and impact of the research.

In terms of contribution to the field of knowledge, the study stands out for three main aspects: First, the creation of an innovative methodology; second, the expansion of the existing foundations in the knowledge base, especially with regard to the usefulness and scientific contribution of the inclusion of people with hearing impairments in the industrial sector; and third, the development of a new methodological approach that can assist in the planning of inclusion actions in large corporations.

The work fills an important gap in the literature, which, until now, has focused on the development of assistive technologies for the social inclusion of hearing-impaired individuals but has barely addressed inclusion in the production environment. This research offers a new and relevant perspective on the inclusion of people with hearing impairments in the context of industrial work.

However, there are limitations to be considered. For example, the instrument was validated with a homogeneous group and a small number of workers, which may compromise the representativeness and generalizability of the results. Furthermore, the study was conducted in a single sector, which limits the application of the findings to other environments or logistics sectors with different characteristics.

In 2025, there was an increase in the hiring of hearing-impaired workers in the logistics sector. The authors intend to continue the study in the future, applying the questionnaire to workers hired over a longer period of time.

## 5. Conclusions

The perception of risks by operators with hearing impairments is essential to expand knowledge about assistive technologies and adaptations in a work environment. Actively listening to workers and their perceptions as reported in a questionnaire makes them feel safer in their work environment.

The results of the study indicate a better understanding of the risks related to work involving interaction between people with hearing impairments and machines, in addition to promoting a safer inclusion of these people in the production environment. The methodology developed proved to be useful and applicable in different industrial environments, also contributing to the advancement of scientific knowledge in the area, which has few specific studies on the subject.

The safety of hearing-impaired workers in logistics environments, especially when interacting with forklift operations, requires a comprehensive approach, involving training, ergonomic adaptations, assistive technologies and legal compliance [24].

This study contributes significantly to the advancement of knowledge in the area of inclusion in the industrial workplace, providing a basis for future investigations and practical actions that promote greater safety and social inclusion.

The future of the study presented in this manuscript involves testing the risk perception questionnaire developed in other sectors, other companies and other industrial branches, since each of the previous ones has its own particularities in terms of environment, training and safety. It is possible that, in different locations, the risk perception questionnaire developed may not reflect the reality of the risks and needs of hearing-impaired workers, therefore requiring adaptation of the questionnaire.

## Figures and Tables

**Figure 1 ijerph-22-00884-f001:**
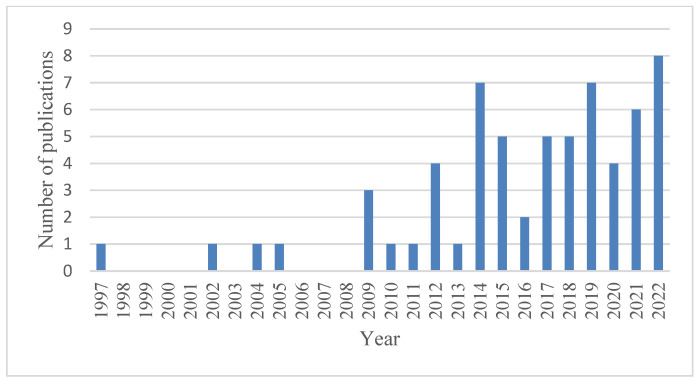
Annual scientific production of articles on hearing-impaired people, inclusion and assistive technologies.

**Figure 2 ijerph-22-00884-f002:**
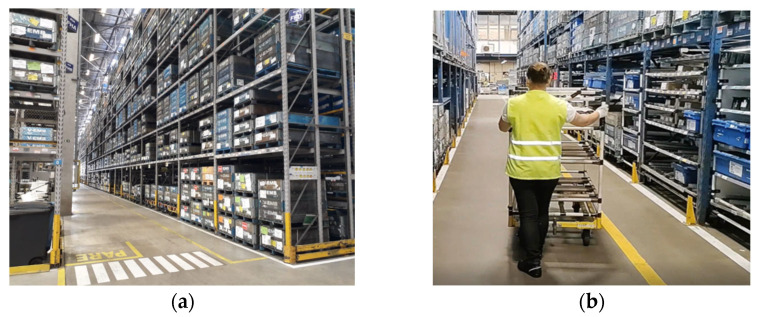
Location of the study. (**a**) caption, (**b**) caption.

**Figure 3 ijerph-22-00884-f003:**
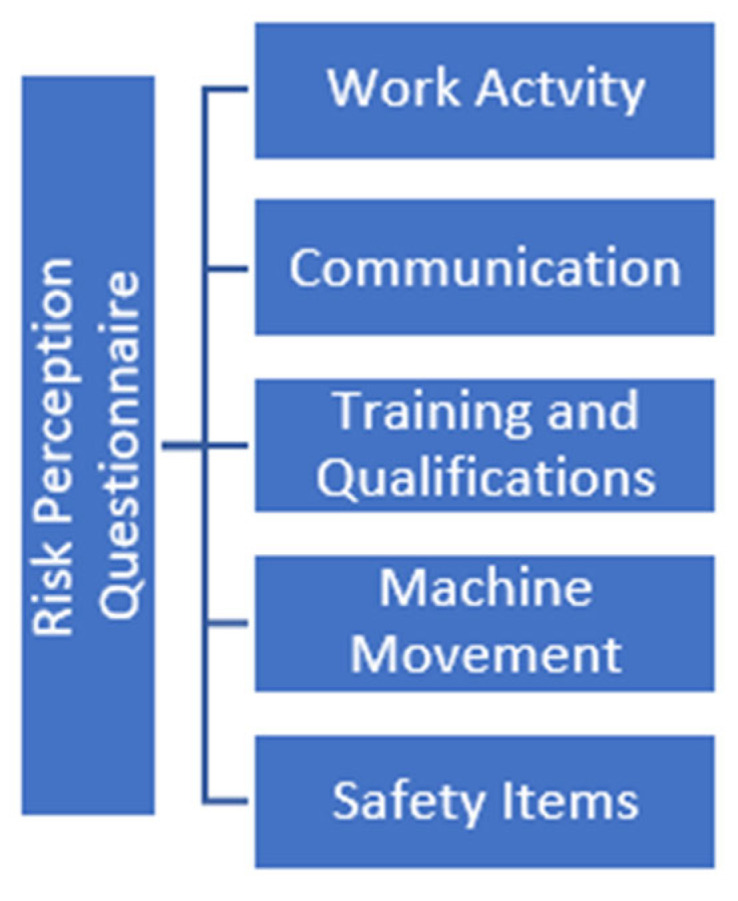
Factors to be evaluated in a productive work environment.

**Figure 4 ijerph-22-00884-f004:**
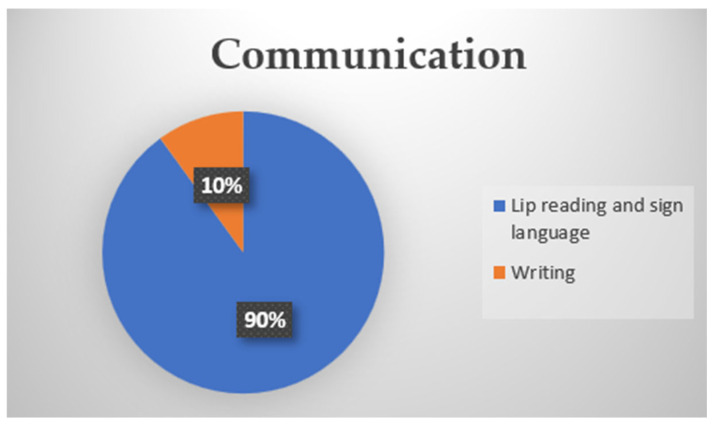
Pie chart for communication.

**Figure 5 ijerph-22-00884-f005:**
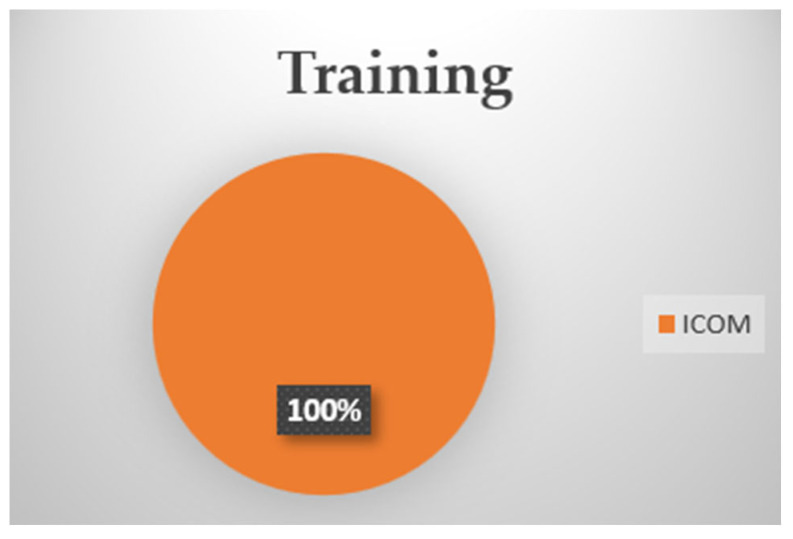
Pie chart for training.

**Figure 6 ijerph-22-00884-f006:**
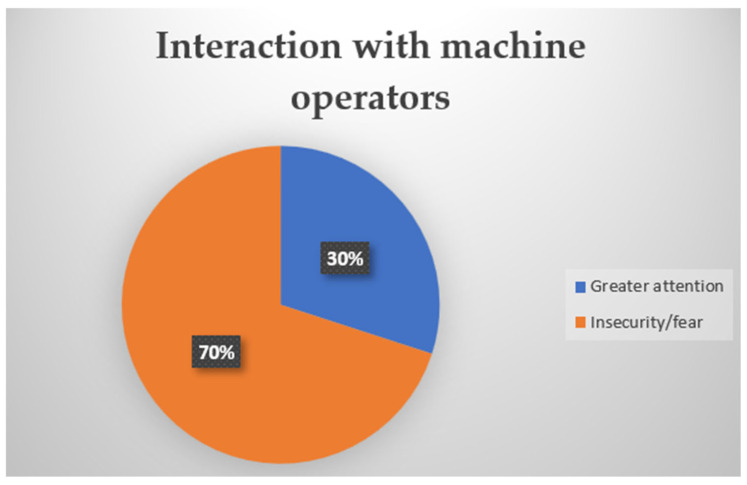
Pie chart for perception of interactions with machine operators.

**Figure 7 ijerph-22-00884-f007:**
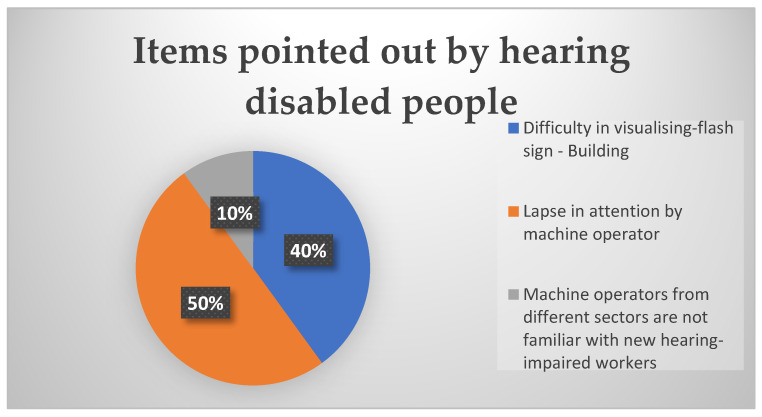
Pie chart for items pointed out by hearing disabled people.

**Table 1 ijerph-22-00884-t001:** Risk Perception Questionnaire.

1. Block 1-General information a. Time with the company: b. What is your level of deafness? Mild Hearing Loss ( ) Moderate Hearing Loss ( ) Severe Hearing Loss ( ) Profound Hearing Loss ( )
2. Block 2-Work activity a. Briefly describe your work activities
______________________________________________________________________________________________________________________________________________________
3. Block 3-Comunication a. What is your best way to communicate? ( ) Writing ( ) Sign Language ( ) Lip Reading b. What is your best way of understanding? ( ) Writing ( ) Sign Language ( ) Lip Reading c. How are work information and operational rules passed on to you? ( ) Written ( ) Presentation ( ) Spoken ( ) Sign Language d. Does your leadership find it easy to communicate with you? ( ) Yes ( ) No e. Is the information conveyed clearly? ( ) Yes ( ) No f. Are you involved in situations to improve your environment/workplace? ( ) Yes ( ) No
4. Block 4-Training g. When you were hired for work, were you given any training regarding specific safety rules in a logistics environment? ( ) Yes ( ) No h. Is there training on operational rules for the logistics sector? ( ) Yes ( ) No i. Are existing training courses updated? ( ) Yes ( ) No
5. Block 5-Interaction with machine operators j. Is there frequent machine operators in your sector? ( ) Yes ( ) No k. Is there machines signage in your sector? ( ) Yes ( ) No l. Can you quickly and easily identify the proximity of these machines? ( ) Yes ( ) No m. How do you feel when working near machines? ( ) Calm ( ) Attentive ( ) Concerned
6. Block 6-Safety items n. Have you ever experienced a situation where there was a safety risk when passing through machines? ( ) Yes ( ) No o. Is there any safety standard to be followed when approaching machines? ( ) Yes ( ) No p. Is there any difficulty in seeing the visual signals? ( ) Yes ( ) No q. What do you think could be improved regarding safety in your work environment? ________________________________________________ ______________________________________

## Data Availability

The data presented in this study are available upon request to the corresponding author because it is an ongoing study and has some restricted information.

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
