# Peer review of "Use of a Validated Risk Perception Questionnaire for the Inclusion of People with Hearing Impairments in a Productive Environment"

_ijerph, 2025, doi:10.3390/ijerph22060884_

Round 1
Reviewer 1 Report
Comments and Suggestions for Authors
In this study, the authors developed a risk perception questionnaire and assessed the barriers and challenges faced by hearing-impaired individuals in logistics environments. This work is valuable for promoting a safer and more efficient workplace. However, the manuscript would benefit from improved clarity, conciseness, and a more formal tone. Below are several questions and suggestions for revision:
- General Refinement
The overall text should be polished for conciseness and to avoid repetitive narration. Below are a few specific examples:- Section 1: Introduction
It is recommended to restructure this section into three concise paragraphs: (1) background, (2) the problem to be addressed, and (3) the main objectives of the current study. The length of this section can be reduced to approximately one-third of its current size. - Section 2: Materials and Methods
The first paragraph (lines 92–95) could be removed, as it adds limited value to the methodology.
- Section 1: Introduction
- Participant Information
Four hearing-impaired individuals participated in the questionnaire. Please clarify whether they used hearing aids. If so, provide information on their corrected hearing ability. If not, from a medical standpoint, should they be advised to use hearing aids first? - Scope and Applicability
Given that only four hearing-impaired participants were included, is the study's outcome intended to be generalized across the industry, or is it specifically tailored to a single company? - Figures
The presentation of figures should be revised for clarity and efficiency:- Figure 1: Consider whether this figure is essential to the manuscript.
- Figure 3: It may be more effectively presented as a concise table.
- Figure 5: Also better suited as a table. Please ensure alignment among the table’s sections for improved readability.
- Table 1
It is recommended to use a three-line table for improved aesthetics and professional presentation. - Results Section
Replace bullet points with Arabic numerals where applicable. Bullet points are generally avoided in formal academic writing.
Author Response
Comments 1: General Refinement: The overall text should be refined to ensure conciseness and avoid repetitive narration.
Answer 1: The text was adapted in general. Thanks for pointing that out, I agree with this comment.
Comments 2: Section 1: Introduction
It is recommended to restructure this section into three concise paragraphs: (1) context, (2) problem to be addressed, and (3) main objectives of the current study. The size of this section can be reduced to approximately one-third of its current size.
Answer 2: I agree. Thanks. The Introduction has been adjusted as requested, with fewer paragraphs and being clearer in citing the context, problem to be addressed and main objectives of the study. These corrections are found in lines 31 to 59.
Comments 3: Section 2: Materials and methods
The first paragraph (lines 92 to 95) could be removed as it adds limited value to the methodology.
Answer 3: In the Materials and Methods Section, lines 92 to 95 were removed as requested.
Comments 4: Participant Information
Four individuals with hearing impairment responded to the questionnaire. Please clarify whether they used hearing aids. If so, please provide information about their corrected hearing ability. If not, from a medical point of view, should they be advised to use hearing aids first?
Answer 4: The public was better informed as requested in lines 120 to 125.
Comments 5: Scope and applicability
Considering that only four participants with hearing impairment were included, is the study result intended to be generalizable to the entire industry or is it specifically tailored to a single company?
Answer 5: Regarding the scope and applicability, it was specified in the development of the study and in the discussion item that this study can be used in other companies that have hearing impaired people in a logistics environment with interaction with mobile machines, such as forklifts.
Comments 6 : Figures
The presentation of figures should be revised for clarity and efficiency:
Figure 1: Consider whether this figure is essential to the manuscript.
Answer Figure 1: I believe that Figure 1 is essential, as it shows the study site. A logistics site, with corridors and human-machine interaction.
Figure 3: Can be presented most effectively as a concise table.
Answer Figure 3: Figure 3 after analysis, was removed from the study.
Figure 5: Also best suited as a table. Make sure the table sections are aligned for easy reading.
Answer Figure 5: Was modified using a table model, with a better explanation of the questions to be asked, line 258 to 259.
Comments 7: Table
It is recommended to use a three-row table to improve aesthetics and professional presentation.
Answer 7: The Table 1, after analysis, was removed from the study and was inserted as text in lines 261 to 266.
Comments 8: Results Section
Replace bullets with Arabic numerals where applicable. Bullets are generally avoided in formal academic texts.
Answer 8: The results were better structured and identified in text format with Arabic numbers on lines 294 to 302.
Reviewer 2 Report
Comments and Suggestions for Authors
Title: you could call this “Development” of …
General-some mild tense confusion, some unnecessary words, some of the language can be shortened to be more clear. Examples given. Some of the terms used are not currently politically correct in the USA (diversity and inclusion), but I would use them anyway. There are many one sentence paragraphs, and a few things I would move for flow purposes and ease of reading.
This is a laudable article on an important topic that is poorly studied to date. The article itself needs more explanation and clarification of methods, validation and reliability.
Abstract
Page 1, line 17 is past tense, so I believe “identifying” in the next line should be past tense as well.
Line 22: What does ‘they’ refer to?
Introduction-can be organized to flow better and you want to avoid one sentence paragraphs. Use the introduction to draw us in...Disabled workers can play a significant role in the workforce and it can benefit them...Ergonomics can help improve safety...The research says...Therefore we created a survey to use to...
Page 1, line 30, I would get rid of the ‘so’ and explain what logistics involves as a job.
Line 31 makes reference to forklifts, but later they are described as mobile machines. I would include that clarification when you first mention forklifts (mobile machines such as forklifts).
Line 35: ‘centres of distribution’ can be simplified…management of large scale distribution centers for shipping and trucking. Is that what a career in logistics is?
Line 37: I fear there is no way to guarantee the safety of workers. You can improve, strengthen, etc, but guarantee, I’m afraid not.
Line 41: sentence should be rearranged so that the definition of ergonomics comes as soon as it is mentioned. I recommend also moving up couple later statements from page 2 to start that paragraph: The insertion of people with disabilities into the labor market is one way to enable the process of accommodating physical, mental, hearing and visual limitations in organizations and society. In addition, it boosts self esteem and helps workers with hearing impairment realise that they are….It should be noted that deafness…training in sign language. Thus, disabled operators face additional challenges when it comes to safety at work, but they can be a valuable part of the workforce and improve their own quality of life.
This could be a new paragraph: Ergonomics is a discipline that seeks to optimize human well-being and system performance and therefore attention to ergonomics can reduce the risk of accidents.
Page 2:
Line 54: I recommend strong verb statements: In addition, implementing auxiliary technology…’’
Line 74: Are there sign language trainings from other countries that also that mitigate limitations? Perhaps just putting “sign language” there?
Page 3:
Line 105: What is ICOM short for?
Line 109: signed a ‘consent’ form?
Lines 112-114 are confusing. Perhaps you want to say it in text:
Four hearing-impaired logistics operators who experienced profound bilateral sensorineural hearing loss, profound hearing loss, severe sensorineural hearing loss or profound bilateral sensorineural hearing loss were initially included in the study. These individuals sat on trollies and worked to create kits of materials in corridors where there was no interaction between man and machine, as these are the safest corridors.
Line 117: I’m confused about the word “validation”-I think what you did was observe the 4 individuals, right? Not validate anything at that point.
Page 4: line 141:How many hours/days did you monitor these 4 people? How many people were observers? How did you get the concepts in Figure 2? Did the observers compare notes with literature review? More explanation needed.
Line 146: Why did you create 6 sets of questions? What does the final survey look like?
Page 5: I’m not sure we need Figure 3 unless we know the questions you ask.
Line 151: What were your search terms? Did you do the lit review before the observation time? Did you search healthcare journals, not just newspapers?
What did you do between lit review and expert panel? How did you develop the questions? Was this from researcher discussion and brainstorming after the observation time and lit review? Did you interview the 4 workers? Some activity is missing between line 154 and line 155.
Line 157-8 is in Portuguese?
Line 159: In order to develop a practical and relevant tool, the judges eliminated four questions for clarity and relevance.
More must be discussed on validity and reliability. Repeated measures must be done to confirm reliability. Therefore please explain how you established reliability.
Are you claiming the survey is valid because it corresponds with established theories or how did you validate it?
Pilot test-how many? Did you do a power calculation?
Focus group-was this done before or after the survey was developed? It seems that it would have been helpful for developing the survey. And did you do multiple focus groups or multiple interviews or both?
Page 6-7
I don’t understand the figures at the bottom of page 6 and top of page 7. They are dark, hard to read, and not explained.
What were the questions asked (is that what Figure 5 is?)? Were they answered on a Likert Scale? Are security items another category of questions? It would be nice to see the survey with the potential answers.
Page 6:
Line 206: past tense: operators who worked in…
Results: very confusing.
First say how many surveys were collected, what kind of sampling (convenience?), how they were analyzed, then the results.
Lines 225 and 235 should be for your discussion, this section is only your findings and what you did with them.
Results concentrates on the findings, such as:
50% of participants felt that additional training would be helpful.
41% felt insecure using the equipment.
Line 291: the authors emphasize (unless it is only one of the authors sited from the reference?)
No mention of limitations or next steps.
The research must be able to be reproduced by a reader.
Comments on the Quality of English Language
Very good, see above, just some confusion with tenses and a few corrections.
Author Response
Comments 1: Quality of the English Language
(x) The English could be improved to express the research more clearly.
Answer 1: I agree, some verb tenses were confusing. The English was revised again by a qualified native speaker.
Comments 1.1: Does the introduction provide sufficient context and include all relevant references?
Answer1.1: Adjustments were made to the introduction, making it more concise, addressing the context, problem to be addressed and main objectives of the current study with a reduction in paragraphs.The research design, methods and results were adjusted.
Comments 2:
Title: You could call this “Development” of… THE ARTICLE APPLIES THE QUESTIONNAIRE, THE DEVELOPMENT HAPPENS AT ANOTHER TIME.
Answer 2: I agree. The title of the article is about the use of a validated risk perception questionnaire for hearing impaired people. I have adjusted the text in general for better understanding.
Comments 3:
General: Some confusion with verb tense, some unnecessary words, some language could be shortened to be clearer. Examples provided. Some of the terms currently used are not politically correct in the US (diversity and inclusion), but I would use them anyway. There are too many one-sentence paragraphs, and some points I would move to make it easier to read and flow.
Answer 3: I agree. Clarity adjustments have been made.
Comments 4:
This is a commendable article on an important and understudied topic to date. The article itself needs more explanation and clarification on methods, validation, and reliability.
Answer 4: Items regarding study initiation have been added for clarity. Clarifications on validation and reliability methods.
Comments 5: Abstract:
Page 1, line 17 is in the past tense, so I believe “identify” on the next line should also be in the past tense.
Answer 5: Verb tense has been adjusted on line 17, page 1.
Comments 6:
Line 22: What does "they" refer to?
Answer 6: The sentence on line 22 has been adjusted to make the sentence more clearly understood.
Comments 7: Introduction
Can be organized to flow better and you want to avoid one-sentence paragraphs. Use the introduction to draw us in... Workers with disabilities can play a significant role in the workforce and this can benefit them... Ergonomics can help improve safety... Research says... So we created a survey to use...
Answer 7: The Introduction has been adjusted as requested, with fewer paragraphs and being clearer in citing the context, problem to be addressed and main objectives of the study. These corrections are found in lines 31 to 59.
Comments 8: Page 1, line 30, I would get rid of the "then" and explain what logistics involves as a job.
Answer 8: The adjustments were made Page 1, line 30.
Comments 9: Line 31 references forklifts, but later they are described as mobile machinery. I would include this clarification when you first mention forklifts (mobile machinery, such as forklifts).
Answer 9: Adjustments were made, explaining in the first sentence, after which it only says forklifts, as it was already explained in line 32.
Comments 10: Line 35: "distribution centers" can be simplified to... management of large distribution centers for transportation and road transport. Is that what a career in logistics is?
Answer 10: Line 25, adjustments were made as requested.
Comments 11: Line 37: I'm afraid there's no way to guarantee the safety of workers. You can improve it, strengthen it, etc., but I'm afraid there's no way to guarantee it.
Answer 11: Line 37, adjustments were made as requested. "Regarding well-being and safety, ergonomics play an essential role in under-standing and improving human interactions with products, equipment, environments and systems aiming to optimize performance, protecting the health and improving the safety and well-being of individuals [2]. "
Comments 12: Line 41: The sentence should be rearranged so that the definition of ergonomics appears immediately after its mention. I also recommend moving forward some later statements on page 2 to begin this paragraph: The inclusion of people with disabilities in the labor market is a way of enabling the process of accommodating physical, mental, auditory and visual limitations in organizations and society. In addition, it increases self-esteem and helps workers with hearing impairments to realize that they are... It should be noted that deafness... sign language training. Thus, operators with disabilities face additional challenges when it comes to workplace safety, but they can be a valuable part of the workforce and improve their own quality of life.
Answer 12: The concept of ergonomics and its objectives was added. "Regarding well-being and safety, ergonomics play an essential role in understanding and improving human interactions with products, equipment, environments and systems aiming to optimize performance, protecting the health and improving the safety and well-being of individuals [2]. Thus, ergonomics becomes a crucial factor, as adaptations in the work environment can contribute to improve safety by reducing the risk of accidents [3,4]." Line 36 to 40.
Comments 13: Page 2: Line 54: I recommend strong verbal statements: In addition, implementing assistive technology…''
Line 74: Are there sign language trainings from other countries that also mitigate the limitations? Maybe just put "sign language" here?
Answer 13: The appropriate adjustments were made as requested, on line 113.
Comments 14: Page 3: Line 105: What is the abbreviation for ICOM?
Answer 14: The term was explained in line 114.
Comments 15: Line 109: Have you signed a 'consent' form?
Answer 15: All operators who participated in the research signed a consent form, line 112. It is an essential document in research involving human beings, where participants freely express their willingness to participate, after having been clearly and in detail informed about all relevant aspects of the research.
Comments 16: Lines 112-114 are confusing. You may want to explain this in text: Initially, four hearing-impaired logistics operators were included in the study, who had profound bilateral sensorineural hearing loss, profound hearing loss, severe sensorineural hearing loss, or profound bilateral sensorineural hearing loss. These individuals, seated on carts, worked on creating material kits in corridors where there was no human-machine interaction, because this was the safest.
Answer 16: The text has been adjusted for better understanding on lines 120 to 125.
Comments 17: Line 117: I'm confused by the word "validation" — I think what you did was observe the 4 individuals, right? Not validate anything at that point.
Answer 17: The observation is correct, it has been adjusted. Initially there was an observation of the activities, lines 138 to 140 are better explained.
Comments 18: Page 4: line 141: How many hours/days did you monitor these 4 people? How many people were observers? How did you obtain the concepts in Figure 2? Did the observers compare their notes with the literature review? More explanation needed.
Answer 18: It was better explained how the observation was carried out, lines 138 to 142.
Comments 19: Line 146: Why did you create 6 sets of questions? What did the final survey look like?
Answer 19: The 6 blocks of questions were created according to the responses to the data collection questionnaire applied to hearing impaired operators and machine operators. Through the responses, it was felt necessary to address these items in the questions of the risk perception questionnaire.
Comments 20: Page 5: I'm not sure we need Figure 3 unless we know the questions you ask.
Answer 20: I removed the figure and added the questions as they were asked to the workers, in a table for better understanding. Line 258 and 259.
Comments 21: Line 151: What were your search terms? Did you do a literature review before the observation period? Did you search health journals, not just newspapers?
Answer 21: The search terms were adjusted, bibliographic review in lines 77 to 113.
Comments 22: What did you do between the literature review and the expert panel? How did you develop the questions? Was this the result of discussion and brainstorming with the researchers after the observation period and the literature review? Did you interview the four workers? Is there any activity missing between lines 154 and 155?
Answer 22: Adjustments were made to lines 138 to 142. The questionnaire validation is found in lines 165 to 243.
Comments 23: Is line 157-8 in Portuguese?
Answer 23: Sorry, fixed the language.
Comments 24: Line 159: In order to develop a practical and relevant tool, the judges eliminated four questions for clarity and relevance. More information on validity and reliability should be discussed. Repeated measures should be conducted to confirm reliability. Therefore, explain how you established reliability.
Answer 24: Corrections were made and the steps and results of the questionnaire validation were inserted in lines 224 to 249.
Comments 25: Are you claiming that the research is valid because it matches established theories or because you validated it?
Answer 25: The risk perception questionnaire has been validated. Lines 168 to 249 show the validation process of this questionnaire.
Comments 26: Pilot test - how many? Did you do a power calculation?
Answer 26: No pilot test was carried out, but rather the tool was used with other hearing impaired people, who at the initial stage were not yet in the company.
Comments 27: Focus group — was this done before or after the research was developed? It seems like it would have been helpful in developing the research. And did you conduct multiple focus groups, multiple interviews, or both?
Answer 27: This part was excluded and explained better in the questionnaire validation, in lines 168 to 249.
Comments 28: What were the questions asked (is this what Figure 5 represents?)? Were they answered on a Likert scale? Are security items another category of questions? It would be interesting to see the survey with the possible answers.
Answer 28: Adjusted Figure 5 to Table 1 for better visualization of the questions. They were answered in closed questions and one in open question. Lines 258 and 259. The main answers are identified in lines 285 to 340.
Comments 29: Page 6: Line 206: past tense: operators who worked in…
Answer 29: Adjusted
Comments 30: Results: Very confusing. First tell how many surveys were collected, what type of sampling (convenience?), how they were analyzed, and then the results.
Answer 30: The writing of the results has been adjusted for better understanding and comprehension. Lines 289 to 340.
Comments 31: Lines 225 and 235 should be for your discussion; this section just contains your findings and what you did with them.
Answer 31: Discussion has been adjusted as requested, lines 342 to 368
Comments 32: The results focus on findings such as: 50% of participants felt additional training would be helpful. 41% felt unsafe using the equipment.
Answer 32: Lines 376 to 412 concentrate the results and recommendations.
Comments 33: No mention of limitations or next steps.
Answer 33: Next steps and limitations on lines 398 to 408.

Round 2
Reviewer 1 Report
Comments and Suggestions for Authors
The revision is much improved, but the full text still needs to be further condensed and focused to emphasize the key points.
Author Response
Comments: The review has been greatly improved, but the full text still needs to be more condensed and focused to emphasize the main points.
Answer: The article has been fully revised and condensed according to the suggestions, focusing on the main points of the study.
Reviewer 2 Report
Comments and Suggestions for Authors
General:
There is a format to scientific papers that must be followed.
Introduction introduces the topic and explains why it is important.
Methods describe what was done so that others can repeat the experiment. Limitations are not included here.
Results tell what was found and must follow what described in Methods
Discussion is how the research impacts the industry, limitations and what the authors intend to do in the future to build on the research.
Conclusion summarizes the main points, restates the thesis, and provides a lasting impression on the reader
Don’t repeat yourself.
Try to avoid “it” or “this”as they decrease the clarity.
All paragraphs have to be more than one or two sentences.
Don’t start sentences with a number of And. Instead say, “A total of 40%...” or something like that.
Introduction:
Line 28-Please put the description of logistics sector here, first thing. Don’t repeat it later.
41-are you supporting or promoting?
The sentence starting on line 41 is too long. I could end it at “logistics environments.” Then, to avoid “this” you can say, “For this purpose, the authors developed a risk perception questionnaire assessing the barriers and challenges faced by the hearing impaired employees at a machinery logistics company.”
Line 55: explain what the significant benefits are that you found in the literature.
Methods:
Line 64: Limitations go in Discussion
Line 68-this should go in line 28.
Line 78-what were your search terms? What did you exclude?
The paragraph starting on line 95 is not needed here, you could put in the Discussion something like: There has been an increasing trend in the number of publications addressing hearing impairment since 2012.
Line 103-What questionnaire was used before you developed yours?
Line 126-133 is repetitive. Please condense.
Line 140-“Forlift operators with whom they interact were observed.
Line 169-you used one survey, then you developed another survey, right?? That description goes here.
Line 189-remove “after evaluation, initially planned to be included in the questionnaire.” As it is repetitive.
Line 190-every measure was required to…(should be past tense)
Line 198-“The questions with the lowest…” Lowest what?
Line 227-242 can be condensed as you don’t need to teach us how to calculate the alpha, just describe that you did it and what it was.
Line 247-“…with a larger hiring audience in 2025.” What does that mean? It is in your future plans to do that? If so, it should go in
Discussion.
Figure 4 is hard to read with the shading, and not totally clear that it connect to the table of question categories
Line 255-Closed-ended questions
Line 259-Quadro should be Figure or Table. And “Item” should be removed before Training in the table.
Results
Lines 270-278 are confusing , I recommend you don’t use the original numbering, but create new numbering for the final survey.
Line 291-Where is Sign Language in Figure 5? Did you combine it with lip reading? Should they both be capitalized?
Lines 294-302 should be in discussion. Also, I'm not sure that you can validate the survey with so few responses, but you describe what you did, so that's satisfactory for me.
Is 'handling of machinery' the same as 'Machine Movement'? Choose one and use it consistently.
Is 'Security Items' the same as 'Safety Items'?
Where are the results for Training and Qualifications?
Line 329, don’t start sentences with a number or And. Instead say, “A total of 40%...” or something like that.
Discussion
Things described earlier.
Conclusion
Line 374- ‘was’ should be ‘is’ (present tense)
Line 379-384-Don’t start sentences with a number of And. Instead say, “A total of 40%...” or something like that.
Line 387-Don’t say “this”. You can say “both” instead.
Line 392-The results of the study expand the understanding of the risks…
Line 409-410-Avoid 'It 'and 'this' without clarification.
Do you want to put in that you would like to test the survey in other industries or companies?
Comments on the Quality of English LanguageDescribed above in Comments and Suggestions for Authors.
Author Response
*Comments General:
There is a format to scientific papers that must be followed.
Introduction introduces the topic and explains why it is important.
Methods describe what was done so that others can repeat the experiment. Limitations are not included here.
Results tell what was found and must follow what described in Methods
Discussion is how the research impacts the industry, limitations and what the authors intend to do in the future to build on the research.
Conclusion summarizes the main points, restates the thesis, and provides a lasting impression on the reader.
*Response General Comments: Thank you very much for the feedback. The adjustments to the text were made as requested.
Comments : Don’t repeat yourself.
Try to avoid “it” or “this”as they decrease the clarity.
All paragraphs have to be more than one or two sentences.
Don’t start sentences with a number of And. Instead say, “A total
of 40%...” or something like that.
Response General Comments: Thank you very much for the feedback. The adjustments to the text were made as requested.
Introduction:
*Line 28-Please put the description of logistics sector here, first thing. Don’t repeat it later.
*Response to the comment: The description of the logistics sector was added and the other paragraph was removed.
*Line 41-are you supporting or promoting?
The sentence starting on line 41 is too long. I could end it at “logistics environments.” Then, to avoid “this” you can say, “For this purpose, the authors developed a risk perception questionnaire assessing the barriers and challenges faced by the hearing impaired employees at a machinery logistics company.”
*Response to the comment: Thank you very much for the feedback. The adjustments to the text were made as requested.
Methods:
*Line 64: Limitations go in Discussion
*Comment: The request has been made.
*Line 68-this should go in line 28.
*Comment: The request has been made.
*Line 78-what were your search terms? What did you exclude?
*Comment: The search and exclusion terms have been added.
*The paragraph starting on line 95 is not needed here, you could put in the Discussion something like: There has been an increasing trend in the number of publications addressing hearing impairment since 2012.
*Comment: The request has been made.
*Line 103-What questionnaire was used before you developed yours?
*Comment: It was explained better that there was a data collection questionnaire applied to the hearing impaired and machine operators who interact on the line.......
*Line 126-133 is repetitive. Please condense.
*Comment: The request has been made.
*Line 140-“Forlift operators with whom they interact were observed.
*Comment: The request has been made.
*Line 169-you used one survey, then you developed another survey, right?? That description goes here.
*Comment: The request has been made.
*Line 189-remove “after evaluation, initially planned to be included in the questionnaire.” As it is repetitive.
*Comment: The request has been made.
*Line 190-every measure was required to…(should be past tense)
*Comment: The request has been made.
*Line 198-“The questions with the lowest…” Lowest what?
Comment: The request was made, it was adjusted to 'a small group'.
*Line 227-242 can be condensed as you don’t need to teach us how to calculate the alpha, just describe that you did it and what it was.
*Comment: The request was made, the calculation was removed.
*Line 247-“…with a larger hiring audience in 2025.” What does that mean? It is in your future plans to do that? If so, it should go in
*Comment: The request was made, it was adjusted for better understanding, explaining that in the year 2025 there was a higher number of hires of hearing-impaired individuals in the logistics sector.
*Linha 140 - “Os operadores de empilhadeira com quem eles interagem foram observados.
*Comment: The sentence was adapted for better understanding.
*Linha 169 - você usou uma pesquisa, depois desenvolveu outra pesquisa, certo? Essa descrição vai aqui.
*Comment: The description was inserted for better understanding.
Discussion
*Figure 4 is hard to read with the shading, and not totally clear that it connect to the table of question categories
*Comment: The request was made, and what was described in the figure was written in text. Excluding the Figure.
*Line 255-Closed-ended questions
*Comment: It was adjusted to “closed questions” as requested.
*Line 259-Quadro should be Figure or Table. And “Item” should be removed before Training in the table.
*Linha 259 - Quadro deve ser Figura ou Tabela. E "Item" deve ser removido antes de Treinamento na tabela.
*Comment: It was adjusted as requested.
Results
*Lines 270-278 are confusing , I recommend you don’t use the original numbering, but create new numbering for the final survey.
*Comment: It was adjusted as requested.
*Line 291-Where is Sign Language in Figure 5? Did you combine it with lip reading? Should they both be capitalized?
Comentário: A Língua de sinais e ............
*Lines 294-302 should be in discussion. Also, I'm not sure that you can validate the survey with so few responses, but you describe what you did, so that's satisfactory for me.
Is 'handling of machinery' the same as 'Machine Movement'?
Choose one and use it consistently.
*Comment: The discussion, the sentence, and the nomenclature have been adjusted.
*Is 'Security Items' the same as 'Safety Items'?
*Comment: The word was adjusted to be more appropriate for the study.
Where are the results for Training and Qualifications?
*Line 329, don’t start sentences with a number or And. Instead say, “A total of 40%...” or something like that.
*Comment: The sentence was adjusted for suitability.
Conclusion
*Line 374- ‘was’ should be ‘is’ (present tense)
*Comment: The verb tense was adjusted as requested.
*Line 379-384-Don’t start sentences with a number of And.Instead say, “A total of 40%...” or something like that.
*Comment: The beginning of the sentence was adjusted as requested.
*Line 387-Don’t say “this”. You can say “both” instead.
*Comment: The spelling has been adjusted.
*Line 392-The results of the study expand the understanding of the risks…
*Comment: The sentence was adjusted as requested.
*Line 409-410-Avoid 'It 'and 'this' without clarification.
*Comment: It was adjusted as requested.
*Do you want to put in that you would like to test the survey in other industries or companies?
*Comment: It was adjusted in the Conclusion and included the desire to develop the study in other industries.
